OBSERVATION

# Cutaneous Surgical Wounds Have Distinct Microbiomes from Intact Skin

Sameer Gupta,[a]* Alexandra J. Poret,[b,c] David Hashemi,[a] Amarachi Eseonu,[a] Sherry H. Yu,[a]§ Jonathan D'Gama,[a] Victor A. Neel,[a] Tami D. Lieberman[b,c,d,e]

[a]Department of Dermatology, MGH, Boston, Massachusetts, USA

[b]Institute for Medical Engineering and Sciences, Massachusetts Institute of Technology, Cambridge, Massachusetts, USA

[c]Department of Civil and Environmental Engineering, Massachusetts Institute of Technology, Cambridge, Massachusetts, USA

[d]Ragon Institute of MIT, MGH, and Harvard, Cambridge, Massachusetts, USA

[e]Broad Institute of MIT and Harvard, Cambridge, Massachusetts, USA

Sameer Gupta and Alexandra J. Poret contributed equally to this work. Author order was chosen randomly.
Victor A. Neel and Tami D. Lieberman contributed equally to this work.

**ABSTRACT** Infections are relatively rare following cutaneous surgical procedures, despite the potential for wound exposure to pathogens both during surgery and throughout the healing process. Although gut commensals are believed to reduce the risk of intestinal infections, an analogous role for skin commensals has not been described. In fact, the microbiome of normally healing surgical skin wounds has not yet been profiled using culture-independent techniques. We characterized the wound microbiome in 53 patients who underwent skin cancer surgery and healed without signs or symptoms of infection. A week after surgery, several bacterial species displayed significant differences in relative abundance when compared to control, nonoperated skin from the same patient. The relative abundance of the most common bacterium found on intact skin, *Cutibacterium acnes*, was reduced in wounds 5-fold. *Staphylococcus aureus*, a frequent cause of postoperative skin infections, was enriched 6.4-fold in clinically noninfected wounds, suggesting active suppression of pathogenicity. Finally, members of the *Corynebacterium* genus were the dominant organism in postoperative wounds, making up 37% of the average wound microbiome. The enrichment of these bacteria in normally healing wounds suggests that they might be capable of providing colonization resistance. Future studies focused on the biological and clinical significance of the wound microbiome may shed light on normal wound healing and potential therapeutic opportunities to mitigate infection risk.

**IMPORTANCE** Commensal bacteria on skin may limit the ability of pathogenic bacteria to cause clinically significant infections. The bacteria on healing acute wounds, which might provide such a protective effect, have not been described using culture-independent approaches in the absence of antibiotics. We compare the microbiome of wounds a week after skin cancer removal surgery with intact skin from the same patient. We find that the potentially pathogenic species *S. aureus* is common on these healing wounds despite the absence of symptoms or signs of infection. We report that bacteria often considered as potential skin probiotics, including *Staphylococcus epidermidis*, do not reach high relative abundance in wound microbiomes. In contrast, specific members of the *Corynebacterium* genus, rarely associated with infections, were significantly enriched in healing wounds compared to intact skin. Future work is needed to see if *Corynebacterium* species or derivatives thereof could be employed to lower the risk of wound infection.

**KEYWORDS** 16S RNA, *Corynebacterium*, human microbiome, infectious disease, skin microbiome

Address correspondence to Victor A. Neel, vneel@mgh.harvard.edu, or Tami D. Lieberman, tami@mit.edu.

*Present address: Sameer Gupta, Department of Dermatology, Brown University, Providence, Rhode Island, USA.

§Present address: Sherry H. Yu, Department of Dermatology, Dermatology and Plastic Surgery Institute, Cleveland Clinic Foundation, Cleveland, Ohio, USA.

The authors declare a conflict of interest. V.A.N., T.D.L., and A.J.P. have filed a provisional patent related to this work. Neel V.A., Lieberman T.D., Poret A.J. 7 January 2022, filing date. US provisional patent application 63/297,365.

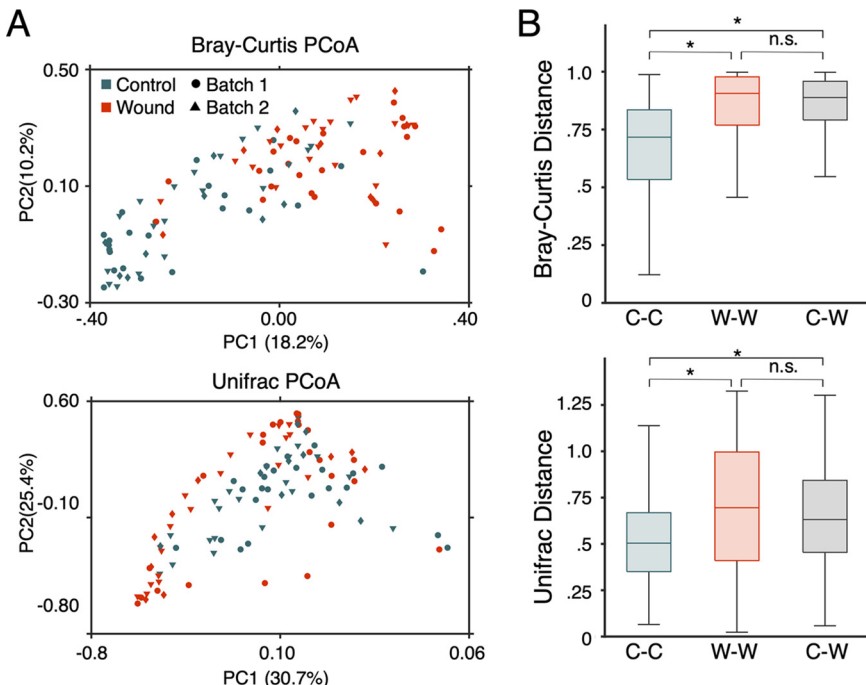

**FIG 1** After surgery, the healthy wound microbiota is disrupted. (A) Bray-Curtis PCoA and UniFrac community composition metrics for both contralateral control (in orange) and wound sites (in blue) after a week postsurgery are shown. Separation between wound and control sites is observed in both composition metrics regardless of sample batch. (B) Comparing the average Bray-Curtis or UniFrac dissimilarity within control samples (labeled C-C), within wound samples (labeled W-W), or between control and wound samples (labeled C-W) displays that microbiome samples from control skin are more similar to one another than wound-normal or wound-wound pairs. *, $P < 10^{-6}$; n.s., not significant.

The structural integrity of skin presents a formidable barrier against invasion by pathogens encountered in the environment. Following the disruption of this barrier—due to surgery, trauma, or other insults—the innate and adaptive arms of the immune system protect against infection until the barrier is reestablished (1). Commensal skin microbes may play an important role in this process and may even provide colonization resistance (2–5), the ability of resident microbiota to mitigate infection risk, akin to what has been described in the gut microbiome (6).

While the microbiome of chronic skin ulcers and burns has been extensively studied in humans (2, 7–9), few human studies have characterized the microbiome in uncomplicated, acute wounds (10, 11) These studies have reported population shifts associated with mechanism of injury and time since wounding (10, 11). However, the administration of broad-spectrum antibiotics prior to sampling may have limited the ability of these studies to identify microbes that colonize normally healing wounds and provide colonization resistance. To our knowledge, the flora that colonize normally healing, uninfected, and nonantibiotic-treated cutaneous wounds has not been described using culture-independent approaches.

Here, we describe the microbiome in normally healing, acute skin wounds following skin cancer surgery. The wound microbiomes of 65 patients undergoing Mohs micrographic surgery and managed by either complete or partial second intention healing were profiled 6 to 8 days after surgery. For each surgical site, an anatomically matched normal, intact skin site was sampled concurrently. The microbiome from each swab sample was profiled using 16S rRNA sequencing of the V1-V3 region, and a custom classifier enabled description of most skin bacteria at the species level (see Methods in the supplemental material) (12, 13). A total of 53 pairs of surgical samples and controls were included in the analysis after quality control (Table S1).

Wounds and anatomically matched control microbiomes had distinct compositions (Fig. 1A). When bacterial composition was visualized in two dimensions using

principal-coordinate analysis (PCoA), wound and control samples clustered separately regardless of anatomical location, cancer type, gender, closure type, or experimental batch (see Fig. S2 to S6 in the supplemental material). Interestingly, wound microbiome compositions showed greater variation across patients than did controls, indicating that the microbiome of wounds can develop in diverse ways (Fig. 1B).

The most striking difference between wounds and control skin was a depletion in the relative abundance of *Cutibacterium*, the most abundant genus in the normal skin microbiome, in wounds (Fig. 2A; $P < 10^{-6}$, Wilcoxon signed-rank). This finding likely reflects the surgical removal of pilosebaceous units in the wound bed, the native niche for this genus (14).

While we did not identify an enrichment of the genus *Staphylococcus* in surgical wounds compared to normal skin microbiomes, significant differences in relative abundances were observed when stratifying the analysis by staphylococcal species (see Table S1). The relative abundances of *Staphylococcus epidermidis* and *Staphylococcus capitis* were lower on wounds relative to normal skin, suggesting that they might not be ecologically successful on these wounds ($P < 0.04$, Wilcoxon signed-rank; Fig. 2B and Table S2). In contrast, *Staphylococcus aureus*, the bacteria most commonly associated with cutaneous wound infections (15), was enriched in surgical sites ($P < 0.002$; Fig. 2B). *S. aureus* was found at ≥5% relative abundance in 30% of healing wound samples, compared to only 11% of normal skin samples. As patients with clinical signs of infection were specifically excluded from this analysis, the high rate of *S. aureus* occupancy in clinically normal wound beds suggests the presence of mechanisms that prevent *S. aureus* pathogenicity.

Wounds were also enriched relative to intact skin in *Corynebacterium* ($P = 0.001$, Wilcoxon signed-rank), a genus primarily composed of species thought to be skin commensals. Since this enrichment could have emerged as an artifact of relative *Cutibacterium* depletion, we accounted for the compositional nature of the data by removing all *Cutibacterium* from our analyses and renormalizing bacterial ratios. After this correction, relative *Corynebacterium* abundance still increased 1.6-fold in surgical wounds, supporting an expansion in the wound niche (Fig. 2B; $P = 0.013$). The species most significantly enriched on wounds was *Corynebacterium tuberculostearicum* ($P < 0.002$) (Fig. 2C), a common, benign commensal (2). While this species has been occasionally been isolated in infections, most of these occurred in immunocompromised patients, and there remains active debate whether the presence of *C. tuberculostearicum* in clinically infected wounds represents the true cause of infection or a contaminant (16, 17). *Corynebacterium accolens*, *Corynebacterium amycolatum*, and *Corynebacterium jeikeium* were also identified in some wounds (Table S2 and Fig. 2C).

The finding of *Corynebacterium* enrichment on normally healing wounds—in the absence of clinical signs of infection—raises the possibility that this genus might help to limit *S. aureus* pathogenicity. A prior study reported a negative correlation between *Corynebacterium* and *S. aureus* relative abundances in the nasal microbiome (18). Similarly, we find a strong negative correlation between these bacterial groups in wounds ($r = -0.55$, Pearson correlation) (see Fig. S7 in the supplemental material); the concordance between nasal and wound environments suggests that *Corynebacterium* can compete with *S. aureus* across niches. *Corynebacterium striatum* has been shown to suppress the *S. aureus agr* virulence pathway *in vitro* (19), providing a possible mechanism for this interaction. In addition, repeated introduction of *Corynebacterium* onto the nasal mucosa of *S. aureus* carriers helped eradicate *S. aureus* colonization in a small human trial (18).

A previous study of traumatic open fracture wounds found only minimal distinction between the microbiome at the wound center and adjacent skin and, in contrast to our findings, depletion of *Corynebacterium* on wounded skin (10). However, patients in this previous study had a different wound type and were treated with broad-spectrum antibiotics prior to sampling, limiting a direct comparison to our work. In one study of chronic wounds, *Corynebacterium* was the only operational

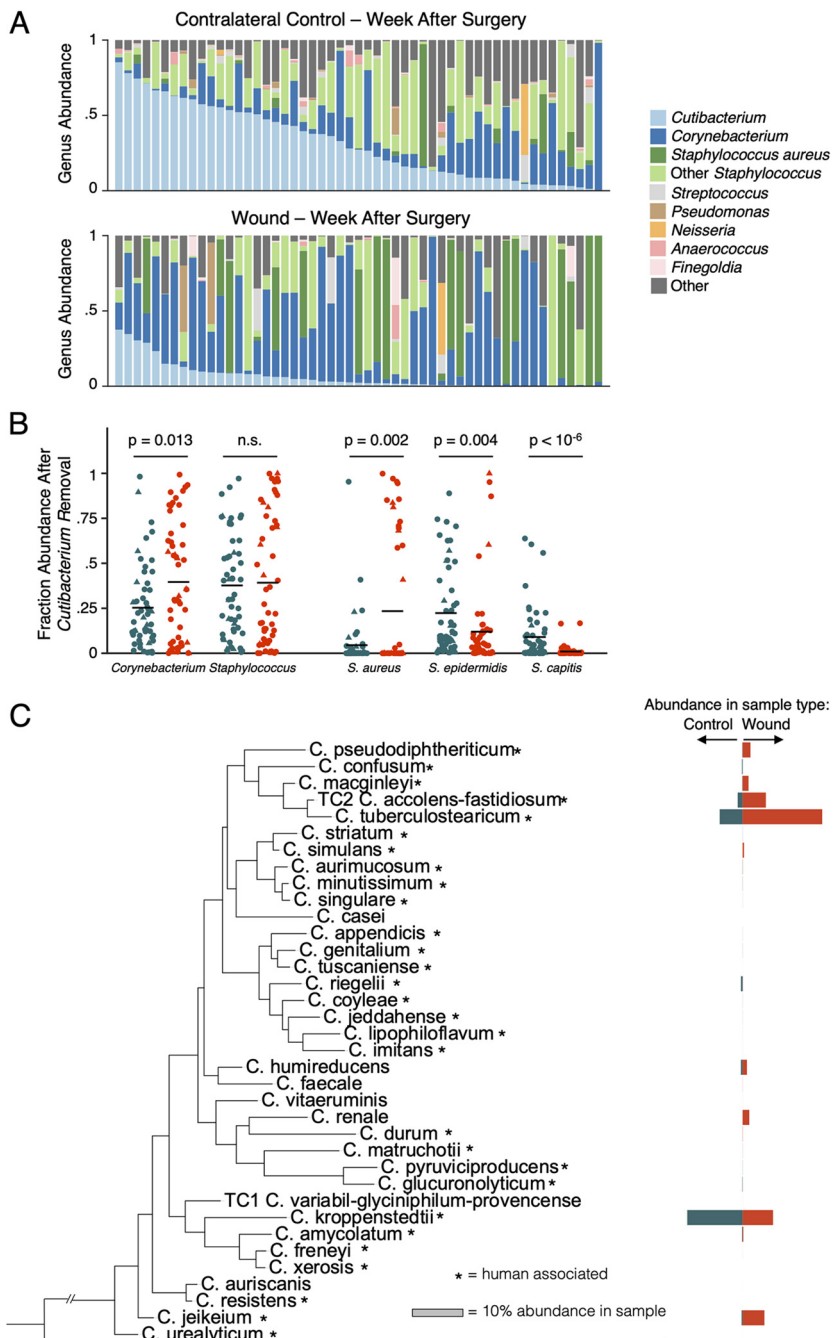

**FIG 2** Specific *Corynebacterium* species are enriched in healthy healing skin wounds. (A) Bar graphs displaying the genus-level composition of all contralateral control and surgery samples sorted by descending *Cutibacterium* abundance visually depicts *Cutibacterium* depletion. When comparing matched control and wound samples, *Cutibacterium* depletion is significant (Wilcoxon rank sum, $P < 10^{-6}$). (B) Plots displaying the composition of various genera and species after the removal of *Cutibacterium* and abundance renormalization are shown for all matched contralateral control and wound samples. Blue dots indicate control samples, while orange dots represent wounds. Batch one samples are additionally marked with circles, and batch two with triangles. *Corynebacterium* is significantly enriched after *Cutibacterium* normalization ($P = 0.013$, Wilcoxon rank sum), while the *Staphylococcus* genus shows no enrichment signal. By breaking apart the *Staphylococcus* genus, *S. aureus* is enriched ($P = 0.002$, Wilcoxon rank sum) while *S. epidermidis* and *S. capitis* are depleted ($P = 0.004$ and $P = 10^{-6}$, Wilcoxon rank sum). (C) A phylogenetic tree created from the 16S rRNA gene of all *Corynebacterium* species observed in matched wound-control samples is shown. Blue bars indicate the average relative abundance observed in control samples, and orange bars indicate wound samples. Asterisks indicate species that have been associate with humans.

taxonomic unit associated with healing, providing additional support for the fitness of *Corynebacterium* on normally healing wounds (20). There are several limitations to the current study. Surgical sites were cleaned with 70% isopropyl alcohol, and some additionally with chlorhexidine, prior to surgery, while control sites were not exposed to anti-infectives. While it has been reported that alcohol and other topical antiseptic treatments temporarily shift the skin microbiome, the influence of these treatments diminishes within hours, and these treatments have been shown to decrease, rather than increase, the relative abundance of *Corynebacterium* (21). Additionally, wound microbiomes may be affected by the presence of cancer-associated microbes (22); however, studies of the cancer microbiome prior to surgery have not detected *Corynebacterium* enrichment (23). Lastly, our sample cohort consisted of primarily elderly patients with extensive sun damage, which may limit the applicability of our findings to other groups.

In conclusion, we observed distinct bacterial communities in acute wounds a week after surgery and anatomically matched normal skin from the same patient. The prevalence of *S. aureus* in these clinically normal wounds was accompanied by an enrichment in the relative abundance of a variety of *Corynebacterium* species. Further work is needed to establish whether wound colonization by *Corynebacterium* or other bacteria plays a role in limiting infection, the specific mechanism underlying this behavior, and if clinicians can leverage this information to prevent of surgical site infections.

**Data availability.** Sequencing data is available under the BioProject number PRJNA809947. Code and data processing scripts can be found at https://github.com/ajporet/cutaneous_wound_microbiome.

## SUPPLEMENTAL MATERIAL

Supplemental material is available online only.
**SUPPLEMENTAL FILE 1**, PDF file, 0.8 MB.

## ACKNOWLEDGMENTS

This research was supported by a Harvard Catalyst grant (to V.A.N.).

V.A.N., T.D.L., and A.J.P. have filed a provisional patent on the use of *Corynebacterium* species for prevention of wound infection.

Conceptualization, V.A.N.; Methodology, S.G., S.H.Y., V.A.N., and T.D.L.; Investigation, S.G., D.H., A.E., and V.A.N.; Data Curation, S.G. and A.J.P.; Formal Analysis, A.J.P. and T.D.L.; Writing, S.G., A.J.P., V.A.N., and T.D.L.; Supervision, V.A.N. and T.D.L.; Funding Acquisition, V.A.N.

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
