## [Reviewer comments · Microbiology Spectrum]

Microbiology Spectrum

Cutaneous surgical wounds have distinct microbiomes from intact skin

Sameer Gupta, Alexandra Poret, David Hashemi, Amarachi Eseonu, Sherry Yu, Jonathan D'Gama, Victor Neel, and Tami Lieberman

Corresponding Author(s): Tami Lieberman, Massachusetts Institute of Technology

Review Timeline:

Submission Date:	August 19, 2022
Editorial Decision:	October 11, 2022
Revision Received:	November 14, 2022
Accepted:	November 29, 2022

Editor: Jan Claesen

Reviewer(s): The reviewers have opted to remain anonymous.

Transaction Report:

DOI: <https://doi.org/10.1128/spectrum.03300-22>

October 11, 2022

Dr. Tami Lieberman
Massachusetts Institute of Technology
Cambridge

Re: Spectrum03300-22 (Cutaneous surgical wounds have distinct microbiomes from intact skin)

Dear Dr. Tami Lieberman:

Thanks for submitting your research to Spectrum. Your manuscript has now been evaluated by two independent Reviewers, who are both very enthusiastic about your work (as am I). The Reviewers raised some minor comments that hopefully should be readily addressable in a text-only revised version of your manuscript. Looking forward to receiving an updated version soon, and please do not hesitate to reach out should you require an extension of the provided timeframe for resubmission.

Thank you for submitting your manuscript to Microbiology Spectrum. As you will see your paper is very close to acceptance. Please modify the manuscript along the lines I have recommended. As these revisions are quite minor, I expect that you should be able to turn in the revised paper in less than 30 days, if not sooner. If your manuscript was reviewed, you will find the reviewers' comments below.

When submitting the revised version of your paper, please provide (1) point-by-point responses to the issues raised by the reviewers as file type "Response to Reviewers," not in your cover letter, and (2) a PDF file that indicates the changes from the original submission (by highlighting or underlining the changes) as file type "Marked Up Manuscript - For Review Only". Please use this link to submit your revised manuscript. Detailed instructions on submitting your revised paper are below.

Link Not Available

Sincerely,

Jan Claesen

Reviewer comments:

Reviewer #1 (Comments for the Author):

Review of "Cutaneous surgical wounds have distinct microbiomes from intact skin"

This work assessed changes in microbial composition of healing surgical wounds and compared them to comparable intact skin sites on 53 patients who underwent Mohs micrographic surgery. While surface antiseptics were used, none of these patients were prescribed antibiotics pre or post-operatively, providing a unique opportunity to observe the changes in a microbial community during wound healing. The resulting data analyses show dramatic changes in the skin microbiome, including relative depletion of *Cutibacterium acnes* and relative enrichment of *Corynebacterium* sp.. In summary, this work has interesting observations on the microbiome of healing, uninfected wounds that is new to the literature. There were some questions and points of clarification for the authors to address

Technical questions

Can relative abundance be used to support the phrase "indicating an expansion in the wound niche"? Is there support that *Corynebacterium* increased in abundance rather than other members of the microbiome reduced in abundance?

Curious why paired-end sequencing was performed but only the forward read was used for analysis.

Grammatical corrections

Main Text: Sentence that starts with "However, the ability of these studies to identify microbes...." Is a bit difficult to read and could benefit from rephrasing.

Reviewer #2 (Comments for the Author):

The manuscript by Gupta et al. is presenting a distinct microbial composition between healing wounds and control skin. They found several differences as well as the potential of *Corynebacterium* for inhibition of colonization of pathogenic bacteria.

The findings seem interesting and sufficient for an observation paper for the Journal.

Preparing Revision Guidelines

Please return the manuscript within 60 days; if you cannot complete the modification within this time period, please contact me. If you do not wish to modify the manuscript and prefer to submit it to another journal, please notify me of your decision immediately so that the manuscript may be formally withdrawn from consideration by Microbiology Spectrum.

Response to reviewers:

We have copy and pasted each reviewer's comments below. **Red text** indicates our changes, corrections, and responses to the reviewer.

Reviewer #1 (Comments for the Author):

Review of "Cutaneous surgical wounds have distinct microbiomes from intact skin"

This work assessed changes in microbial composition of healing surgical wounds and compared them to comparable intact skin sites on 53 patients who underwent Mohs micrographic surgery. While surface antiseptics were used, none of these patients were prescribed antibiotics pre or post-operatively, providing a unique opportunity to observe the changes in a microbial community during wound healing. The resulting data analyses show dramatic changes in the skin microbiome, including relative depletion of *Cutibacterium acnes* and relative enrichment of *Corynebacterium* sp.. In summary, this work has interesting observations on the microbiome of healing, uninfected wounds that is new to the literature. There were some questions and points of clarification for the authors to address

We thank the referee for their comments on the impact of the manuscript and constructive criticisms.

Technical questions

Can relative abundance be used to support the phrase "indicating an expansion in the wound niche"? Is there support that *Corynebacterium* increased in abundance rather than other members of the microbiome reduced in abundance?

Thank you for this important point. We have changed this sentence to say "supporting" and have made several small modifications throughout to ensure that we refer to "relative abundance" rather than "abundance" throughout.

Curious why paired-end sequencing was performed but only the forward read was used for analysis.

Read quality was not high enough towards the ends of reads to overlap paired reads with high certainty and accuracy. We therefore analyzed only the forward read, which provided ample information for species-level classification of ASVs. We have added this context to the Methods, thank you.

Grammatical corrections

Main Text: Sentence that starts with "However, the ability of these studies to identify microbes...." Is a bit difficult to read and could benefit from rephrasing.

Thank you for noticing this problem. We have fixed the sentence, reproduce below:

“However, the administration of broad spectrum antibiotics prior to sampling may have limited the ability of these studies to identify microbes that colonize normally healing wounds and provide colonization resistance. “

Reviewer #2 (Comments for the Author):

The manuscript by Gupta et al. is presenting a distinct microbial composition between healing wounds and control skin. They found several differences as well as the potential of *Corynebacterium* for inhibition of colonization of pathogenic bacteria.

The findings seems interesting and sufficient for Observation paper of the Journal.

We thank the referee for their positive review of the manuscript.

November 29, 2022

Dr. Tami Lieberman
Massachusetts Institute of Technology
Cambridge

Re: Spectrum03300-22R1 (Cutaneous surgical wounds have distinct microbiomes from intact skin)

Dear Dr. Tami Lieberman:

Thanks for addressing the Reviewers' comments in your revised manuscript! I'd like to congratulate you on the acceptance of your paper for publication in Spectrum!

Your manuscript has been accepted, and I am forwarding it to the ASM Journals Department for publication. You will be notified when your proofs are ready to be viewed.

Sincerely,

Jan Claesen
Editor, Microbiology Spectrum
